# Gated-Attention Readers for Text Comprehension

## Abstract

In this paper we study the problem of answering cloze-style questions over documents. Our model, the Gated-Attention (GA) Reader, integrates a multi-hop architecture with a novel attention mechanism, which is based on multiplicative interactions between the query embedding and the intermediate states of a recurrent neural network document reader. This enables the reader to build query-specific representations of tokens in the document for accurate answer selection. The GA Reader obtains state-of-the-art results on three benchmarks for this task–the CNN & Daily Mail news stories and the Who Did What dataset. The effectiveness of multiplicative interaction is demonstrated by an ablation study, and by comparing to alternative compositional operators for implementing the gated-attention.

## 1 Introduction

A recent trend to measure progress towards machine reading is to test a system's ability to answer questions about a document it has to comprehend. Towards this end, several large-scale datasets of cloze-style questions over a context document have been introduced recently, which allow the training of supervised machine learning systems (Hermann et al., 2015; Hill et al., 2015; Onishi et al., 2016). Such datasets can be easily constructed automatically and the unambiguous nature of their queries provides an objective benchmark to measure a system's performance at text comprehension.

Deep learning models have been shown to outperform traditional shallow approaches on text comprehension tasks (Hermann et al., 2015). The success of many recent models can be attributed primarily to two factors: (1) *Multi-hop architectures* allow a (Weston et al., 2014; Sordoni et al., 2016; Shen et al., 2016), model to scan the document and the question iteratively for multiple passes. (2) *Attention mechanisms*, (Chen et al., 2016; Hermann et al., 2015) borrowed from the machine translation literature (Bahdanau et al., 2014), allow the model to focus on appropriate subparts of the context document. Intuitively, the multi-hop architecture allows the reader to incrementally refine token representations, and the attention mechanism re-weights different parts in the document according to their relevance to the query.

The effectiveness of multi-hop reasoning and attentions have been explored orthogonally so far in the literature. In this paper, we focus on combining both in a complementary manner, by designing a novel attention mechanism which gates the evolving token representations across hops. More specifically, unlike existing models where the query attention is applied either token-wise (Hermann et al., 2015; Kadlec et al., 2016; Chen et al., 2016; Hill et al., 2015) or sentence-wise (Weston et al., 2014; Sukhbaatar et al., 2015) to allow weighted aggregation, the Gated-Attention (GA) module proposed in this work allows the query to directly interact with each dimension of the token embeddings at the semantic-level, and is applied layer-wise as information filters during the multi-hop representation learning process. Such a fine-grained attention enables our model to learn conditional token representations w.r.t. the given question, leading to accurate answer selections.

We show in our experiments that the proposed GA reader, despite its relative simplicity, consistently improves over a variety of strong baselines on three benchmark datasets Our key contribution, the GA module, provides a significant im-

provement for large datasets. Qualitatively, visualization of the attentions at intermediate layers of the GA reader shows that in each layer the GA reader attends to distinct salient aspects of the query which help in determining the answer.

## 2 Related Work

The cloze-style QA task involves tuples of the form $(d, q, a, \mathcal{C})$, where $d$ is a document (context), $q$ is a query over the contents of $d$, in which a phrase is replaced with a placeholder, and $a$ is the answer to $q$, which comes from a set of candidates $\mathcal{C}$. In this work we consider datasets where each candidate $c \in \mathcal{C}$ has at least one token which also appears in the document. The task can then be described as: given a document-query pair $(d, q)$, find $a \in \mathcal{C}$ which answers $q$. Below we provide an overview of representative neural network architectures which have been applied to this problem.

*LSTMs with Attention:* Several architectures introduced in (Hermann et al., 2015) employ LSTM units to compute a combined document-query representation $g(d, q)$, which is used to rank the candidate answers. These include the **DeepLSTM Reader** which performs a single forward pass through the concatenated *(document, query)* pair to obtain $g(d, q)$; the **Attentive Reader** which first computes a document vector $d(q)$ by a weighted aggregation of words according to attentions based on $q$, and then combines $d(q)$ and $q$ to obtain their joint representation $g(d(q), q)$; and the **Impatient Reader** where the document representation is built incrementally. The architecture of the Attentive Reader has been simplified recently in **Stanford Attentive Reader**, where shallower recurrent units were used with a bilinear form for the query-document attention (Chen et al., 2016).

*Attention Sum:* The **Attention-Sum (AS) Reader** (Kadlec et al., 2016) uses two bi-directional GRU networks (Cho et al., 2014) to encode both $d$ and $q$ into vectors. A probability distribution over the entities in $d$ is obtained by computing dot products between $q$ and the entity embeddings and taking a softmax. Then, an aggregation scheme named *pointer-sum attention* is further applied to sum the probabilities of the same entity, so that frequent entities the document will be favored compared to rare ones. Building on the AS Reader, the **Attention-over-Attention (AoA) Reader** (Cui et al., 2016) introduces a two-way attention mechanism where the query and the doc-

ument are mutually attentive to each other.

*Mulit-hop Architectures:* **Memory Networks (MemNets)** were proposed in (Weston et al., 2014), where each sentence in the document is encoded to a memory by aggregating nearby words. Attention over the memory slots given the query is used to compute an overall memory and to renew the query representation over multiple iterations, allowing certain types of reasoning over the salient facts in the memory and the query. **Neural Semantic Encoders (NSE)** (Munkhdalai & Yu, 2016a) extended MemNets by introducing a *write* operation which can evolve the memory over time during the course of reading. Iterative reasoning has been found effective in several more recent models, including the **Iterative Attentive Reader** (Sordoni et al., 2016) and **ReasoNet** (Shen et al., 2016). The latter allows dynamic reasoning steps and is trained with reinforcement learning.

Other related works include **Dynamic Entity Representation network (DER)** (Kobayashi et al., 2016), which builds dynamic representations of the candidate answers while reading the document, and accumulates the information about an entity by max-pooling; **EpiReader** (Trischler et al., 2016) consists of two networks, where one proposes a small set of candidate answers, and the other reranks the proposed candidates conditioned on the query and the context; **Bi-Directional Attention Flow network (BiDAF)** (Seo et al., 2016) adopts a multi-stage hierarchical architecture along with a flow-based attention mechanism; (Bajgar et al., 2016) showed a 10% improvement on the CBT corpus (Hill et al., 2015) by training the AS Reader on an augmented training set of about 14 million examples, making a case for the community to exploit data abundance. The focus of this paper, however, is on designing models which exploit the available data efficiently.

## 3 Gated-Attention Reader

Our proposed GA readers perform multiple hops over the document (context), similar to the Memory Networks architecture (Sukhbaatar et al., 2015). Multi-hop architectures mimic the multi-step comprehension process of human readers, and have shown promising results in several recent models for text comprehension (Sordoni et al., 2016; Kumar et al., 2015; Shen et al., 2016). The contextual representations in GA readers, namely the embeddings of words in the document, are it-

eratively refined across hops until reaching a final attention-sum module (Kadlec et al., 2016) which maps the contextual representations in the last hop to a probability distribution over candidate answers.

The attention mechanism has been introduced recently to model human focus, leading to significant improvement in machine translation and image captioning (Bahdanau et al., 2014; Mnih et al., 2014). In reading comprehension tasks, ideally, the semantic meanings carried by the contextual embeddings should be aware of the query across hops. As an example, human readers are able to keep the question in mind during multiple passes of reading, to successively mask away information irrelevant to the query. However, existing neural network readers are restricted to either attend to tokens (Hermann et al., 2015; Chen et al., 2016) or entire sentences (Weston et al., 2014), with the assumption that certain sub-parts of the document are more important than others. In contrast, we propose a finer-grained model which attends to components of the semantic representation being built up by the GRU. The new attention mechanism, called *gated-attention*, is implemented via *multiplicative* interactions between the query and the contextual embeddings, and is applied per hop to act as fine-grained information filters during the multi-step reasoning. The filters weigh individual components of the vector representation of *each* token in the document separately.

The design of gated-attention layers is motivated by the effectiveness of multiplicative interaction among vector-space representations, e.g., in various types of recurrent units (Hochreiter & Schmidhuber, 1997; Wu et al., 2016) and in relational learning (Yang et al., 2014; Kiros et al., 2014). While other types of compositional operators are possible, such as concatenation or addition (Mitchell & Lapata, 2008), we find that multiplication has strong empirical performance (section 4.3), where query representations naturally serve as information filters across hops.

## 3.1 Model Details

Several components of the model use a Gated Recurrent Unit (GRU) (Cho et al., 2014) which maps an input sequence $X = [x_1, x_2, \ldots, x_T]$ to an ouput sequence $H = [h_1, h_2, \ldots, h_T]$ as follows:

$$
\begin{aligned}
r_t &= \sigma(W_r x_t + U_r h_{t-1} + b_r), \\
z_t &= \sigma(W_z x_t + U_z h_{t-1} + b_z), \\
\tilde{h}_t &= \tanh(W_h x_t + U_h(r_t \odot h_{t-1}) + b_h), \\
h_t &= (1 - z_t) \odot h_{t-1} + z_t \odot \tilde{h}_t.
\end{aligned}
$$

where $\odot$ denotes the Hadamard product or the element-wise multiplication. $r_t$ and $z_t$ are called the *reset* and *update* gates respectively, and $\tilde{h}_t$ the *candidate output*. A Bi-directional GRU (Bi-GRU) processes the sequence in both forward and backward directions to produce two sequences $[h_1^f, h_2^f, \ldots, h_T^f]$ and $[h_1^b, h_2^b, \ldots, h_T^b]$, which are concatenated at the output

$$
\overleftrightarrow{\text{GRU}}(X) = [h_1^f \| h_T^b, \ldots, h_T^f \| h_1^b] \qquad (1)
$$

where $\overleftrightarrow{\text{GRU}}(X)$ denotes the *full output* of the Bi-GRU obtained by concatenating each forward state $h_i^f$ and backward state $h_{T-i+1}^b$ at step $i$ given the input $X$. Note $\overleftrightarrow{\text{GRU}}(X)$ is a matrix in $\mathbb{R}^{2n_h \times T}$ where $n_h$ is the number of hidden units in GRU.

Let $X^{(0)} = [x_1^{(0)}, x_2^{(0)}, \ldots x_{|D|}^{(0)}]$ denote the token embeddings of the document, which are also inputs at layer 1 for the document reader below, and $Y = [y_1, y_2, \ldots y_{|Q|}]$ denote the token embeddings of the query. Here $|D|$ and $|Q|$ denote the document and query lengths respectively.

### 3.1.1 Multi-Hop Architecture

Fig. 1 illustrates the Gated-Attention (GA) reader. The model reads the document and the query over $K$ horizontal layers, where layer $k$ receives the contextual embeddings $X^{(k-1)}$ of the document from the previous layer. The document embeddings are transformed by taking the full output of a document Bi-GRU (indicated in blue in Fig. 1):

$$
D^{(k)} = \overleftrightarrow{\text{GRU}}_D^{(k)}(X^{(k-1)}) \qquad (2)
$$

At the same time, a layer-specific query representation is computed as the full output of a separate query Bi-GRU (indicated in green in Figure 1):

$$
Q^{(k)} = \overleftrightarrow{\text{GRU}}_Q^{(k)}(Y) \qquad (3)
$$

Next, *Gated-Attention* is applied to $D^{(k)}$ and $Q^{(k)}$ to compute inputs for the next layer $X^{(k)}$.

$$
X^{(k)} = \text{GA}(D^{(k)}, Q^{(k)}) \qquad (4)
$$

where GA is defined in the following subsection.

Figure 1: Gated-Attention Reader. Dashed lines represent dropout connections.

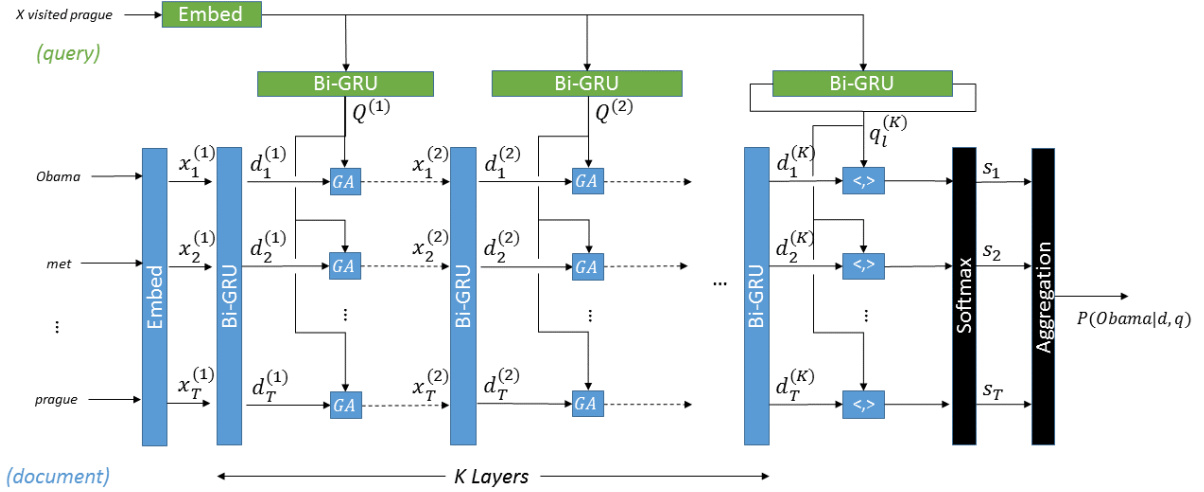

### 3.1.2 Gated-Attention Module

For brevity, let us drop the superscript $k$ in this subsection as we are focusing on a particular layer. For each token $d_i$ in $D$, the GA module forms a token-specific representation of the query $\tilde{q}_i$ using soft attention, and then multiplies the query representation element-wise with the document token representation. Specifically, for $i = 1, \ldots, |D|$:

$$\alpha_i = \text{softmax}(Q^\top d_i) \qquad (5)$$
$$\tilde{q}_i = Q\alpha_i$$
$$x_i = d_i \odot \tilde{q}_i \qquad (6)$$

In equation (6) we use the multiplication operator to model the interactions between $d_i$ and $\tilde{q}_i$. In the experiments section, we also report results for other choices of gating functions, including addition $x_i = d_i + \tilde{q}_i$ and concatenation $x_i = d_i \| \tilde{q}_i$.

### 3.1.3 Answer Prediction

Let $q_\ell^{(K)} = q_\ell^f \| q_{T-\ell+1}^b$ be an intermediate output of the final layer query Bi-GRU at the location $\ell$ of the cloze token in the query, and $D^{(K)} = \overset{\longleftrightarrow}{\text{GRU}}_D^{(K)}(X^{(K-1)})$ be the full output of final layer document Bi-GRU. To obtain the probability that a particular token in the document answers the query, we take an inner-product between these two, and pass through a softmax layer:

$$s = \text{softmax}((q_\ell^{(K)})^T D^{(K)}) \qquad (7)$$

where vector $s$ defines a probability distribution over the $|D|$ tokens in the document. The probability of a particular candidate $c \in \mathcal{C}$ as being the answer is then computed by aggregating the probabilities of all document tokens which appear in $c$ and renormalizing over the candidates:

$$\Pr(c|d, q) \propto \sum_{i \in \mathbb{I}(c,d)} s_i \qquad (8)$$

where $\mathbb{I}(c, d)$ is the set of positions where a token in $c$ appears in the document $d$. This aggregation operation is the same as the *pointer sum attention* applied in the AS Reader (Kadlec et al., 2016).

Finally, the candidate with maximum probability is selected as the predicted answer:

$$a^* = \text{argmax}_{c \in \mathcal{C}} \ \Pr(c|d, q). \qquad (9)$$

During the training phase, model parameters of GA are updated w.r.t. a cross-entropy loss between the predicted probabilities and the true answers.

### 3.1.4 Further Enhancements

*Character-level Embeddings*: Given a token $w$ from the document or query, its vector space representation is computed as $x = L(w) \| C(w)$. $L(w)$ retrieves the word-embedding for $w$ from a lookup table $L \in \mathbb{R}^{|V| \times n_l}$, whose rows hold a vector for each unique token in the vocabulary. We also utilize a character composition model $C(w)$ which generates an orthographic embedding of the token. Such embeddings have been previously shown to be helpful for tasks like Named Entity Recognition (Yang et al., 2016) and dealing with OOV tokens at test time (Dhingra et al., 2016). The embedding $C(w)$ is generated by taking the final outputs $z_{n_c}^f$ and $z_{n_c}^b$ of a Bi-GRU applied to embeddings from

a lookup table of characters in the token, and applying a linear transformation:

$$z = z_{n_c}^f || z_{n_c}^b$$
$$C(w) = Wz + b$$

*Question Evidence Common Word Feature (qe-comm)*: (Li et al., 2016) recently proposed a simple token level indicator feature which significantly boosts reading comprehension performance in some cases. For each token in the document we construct a one-hot vector $f_i \in \{0, 1\}^2$ indicating its presence in the query. It can be incorporated into the GA reader by assigning a feature lookup table $F \in \mathbb{R}^{n_F \times 2}$ (we use $n_F = 2$), taking the feature embedding $e_i = f_i^T F$ and appending it to the inputs of the last layer document BiGRU as, $x_i^{(K)} || f_i$ for all $i$. We conducted several experiments both with and without this feature and observed some interesting trends, which are discussed below. Henceforth, we refer to this feature as the *qe-comm feature* or just *feature*.

## 4 Experiments and Results

### 4.1 Datasets

We evaluate the GA reader on five large-scale datasets recently proposed in the literature. The first two, CNN and Daily Mail news stories[1] consist of articles from the popular CNN and Daily Mail websites (Hermann et al., 2015). A query over each article is formed by removing an entity from the short summary which follows the article. Further, entities within each article were anonymized to make the task purely a comprehension one. N-gram statistics, for instance, computed over the entire corpus are no longer useful in such an anonymized corpus.

The next two datasets are formed from two different subsets of the Children's Book Test (CBT)[2] (Hill et al., 2015). Documents consist of 20 contiguous sentences from the body of a popular children's book, and queries are formed by deleting a token from the 21st sentence. We only focus on subsets where the deleted token is either a common noun (CN) or named entity (NE) since simple language models already give human-level performance on the other types (cf. (Hill et al., 2015)).

The final dataset is Who Did What[3] (WDW) (Onishi et al., 2016), constructed from the LDC English Gigaword newswire corpus. First, article pairs which appeared around the same time and with overlapping entities are chosen, and then one article forms the document and a cloze query is constructed from the other. Missing tokens are always person named entities. Questions which are easily answered by simple baselines are filtered out, to make the task more challenging. There are two versions of the training set—a small but focused "Strict" version and a large but noisy "Relaxed" version. We report results on both settings which share the same validation and test sets. Statistics of all the datasets used in our experiments are summarized in the Appendix (Table 4).

### 4.2 Performance Comparison

Tables 1 and 3 show a comparison of the performance of GA Reader with previously published results on WDW and CNN, Daily Mail, CBT datasets respectively. The numbers reported for GA Reader are for single best models, though we compare to both ensembles and single models from prior work. GA Reader-- refers to an earlier version of the model, unpublished but described in a preprint, with the following differences—(1) it does not utilize token-specific attentions within the GA module, as described in equation (5), (2) it does not use a character composition model, (3) it is initialized with word embeddings pretrained on the corpus itself rather than GloVe. A detailed analysis of these differences is studied in the next section. Here we present 4 variants of the latest GA Reader, using combinations of whether the qe-comm feature is used (+feature) or not, and whether the word lookup table $L(w)$ is updated during training or fixed to its initial value. Other hyperparameters are listed in Appendix A.

Interestingly, we observe that feature engineering leads to significant improvements for WDW and CBT datasets, but not for CNN and Daily Mail datasets. We note that anonymization of the latter datasets means that there is already some feature engineering (it adds hints about whether a token is an entity), and these are much larger than the other four. In machine learning it is common to see the effect of feature engineering diminish with increasing data size. Similarly, fixing the word embeddings provides an improvement for the WDW and CBT, but not for CNN and Daily Mail. This is not surprising given that the latter datasets are larger and less prone to overfitting.

---

[1] https://github.com/deepmind/rc-data
[2] http://www.thespermwhale.com/jaseweston/babi/CBTest.tgz
[3] https://tticnlp.github.io/who_did_what/

Table 1: Validation/Test accuracy (%) on WDW dataset for both "Strict" and "Relaxed" settings. Results with "†" are cf previously published works.

| Model | Strict | | Relaxed | |
|---|---|---|---|---|
| | Val | Test | Val | Test |
| Human † | – | 84 | – | – |
| Attentive Reader † | – | 53 | – | 55 |
| AS Reader † | – | 57 | – | 59 |
| Stanford AR † | – | 64 | – | 65 |
| NSE † | 66.5 | 66.2 | 67.0 | 66.7 |
| GA-- † | – | 57 | – | 60.0 |
| GA (update $L(w)$) | 67.8 | 67.0 | 67.0 | 66.6 |
| GA (fix $L(w)$) | 68.3 | 68.0 | 69.6 | 69.1 |
| GA (+feature, update $L(w)$) | 70.1 | 69.5 | 70.9 | 71.0 |
| GA (+feature, fix $L(w)$) | **71.6** | **71.2** | **72.6** | **72.6** |

Table 2: **Top:** Performance of different gating functions. **Bottom:** Effect of varying the number of hops $K$. Results on WDW without using the qe-comm feature and with fixed $L(w)$.

| Gating Function | Accuracy | |
|---|---|---|
| | Val | Test |
| Sum | 64.9 | 64.5 |
| Concatenate | 64.4 | 63.7 |
| Multiply | **68.3** | **68.0** |
| **K** | | |
| 1 (AS) † | – | 57 |
| 2 | 65.6 | 65.6 |
| 3 | **68.3** | 68.0 |
| 4 | **68.3** | **68.2** |

Comparing with prior work, on the WDW dataset the basic version of the GA Reader outperforms all previously published models when trained on the Strict setting. By adding the qe-comm feature the performance increases by 3.2% and 3.5% on the Strict and Relaxed settings respectively to set a new state of the art on this dataset. On the CNN and Daily Mail datasets the GA Reader leads to an improvement of 3.2% and 4.3% respectively over the best previous single models. They also outperform previous ensemble models, setting a new state of that art for both datasets. For CBT-NE, GA Reader with the qe-comm feature outperforms all previous single and ensemble models except the AS Reader trained on the much larger BookTest Corpus (Bajgar et al., 2016). Lastly, on CBT-CN the GA Reader with the qe-comm feature outperforms all previously published single models except the NSE, and AS Reader trained on a larger corpus.

### 4.3 GA Reader Analysis

In this section we do an ablation study to see the effect of Gated Attention. We compare the GA Reader as described here to a model which is exactly the same in all aspects, except that it passes document embeddings $D^{(k)}$ in each layer directly to the inputs of the next layer without using the GA module. In other words $X^{(k)} = D^{(k)}$ for all $k > 0$. This model ends up using only one query GRU at the output layer for selecting the answer from the document. We compare these two variants both with and without the qe-comm feature on CNN and WDW datasets for three subsets of

the training data - 50%, 75% and 100%. Test set accuracies for these settings are shown in Figure 2. On CNN when tested without feature engineering, we observe that GA provides a significant boost in performance compared to without GA. When tested with the feature it still gives an improvement, but the improvement is significant only with 100% training data. On WDW-Strict, which is a third of the size of CNN, without the feature we see an improvement when using GA versus without using GA, which becomes significant as the training set size increases. When tested with the feature on WDW, for a small data size without GA does better than with GA, but as the dataset size increases they become equivalent. We conclude that GA provides a boost in the absence of feature engineering, or as the training set size increases.

Next we look at the question of how to gate intermediate document reader states from the query, i.e. what operation to use in equation 6. Table 2 (top) shows the performance on WDW dataset for three common choices – sum ($x = d + q$), concatenate ($x = d\|q$) and multiply ($x = d \odot q$). Empirically we find element-wise multiplication does significantly better than the other two, which justifies our motivation to "filter" out document features which are irrelevant to the query.

At the bottom of Table 2 we show the effect of varying the number of hops $K$ of the GA Reader on the final performance. We note that for $K = 1$, our model is equivalent to the AS Reader without any GA modules. We see a steep and steady rise in accuracy as the number of hops is increased from $K = 1$ to 3, which remains constant beyond

Table 3: Validation/Test accuracy (%) on CNN, Daily Mail and CBT. Results marked with "†" are cf previously published works. Results marked with "‡" were obtained by training on a larger training set. Best performance on standard training sets is in bold, and on larger training sets in italics.

| Model | CNN | | Daily Mail | | CBT-NE | | CBT-CN | |
|---|---|---|---|---|---|---|---|---|
| | Val | Test | Val | Test | Val | Test | Val | Test |
| Humans (query) † | – | – | – | – | – | 52.0 | – | 64.4 |
| Humans (context + query) † | – | – | – | – | – | 81.6 | – | 81.6 |
| LSTMs (context + query) † | – | – | – | – | 51.2 | 41.8 | 62.6 | 56.0 |
| Deep LSTM Reader † | 55.0 | 57.0 | 63.3 | 62.2 | – | – | – | – |
| Attentive Reader † | 61.6 | 63.0 | 70.5 | 69.0 | – | – | – | – |
| Impatient Reader † | 61.8 | 63.8 | 69.0 | 68.0 | – | – | – | – |
| MemNets † | 63.4 | 66.8 | – | – | 70.4 | 66.6 | 64.2 | 63.0 |
| AS Reader † | 68.6 | 69.5 | 75.0 | 73.9 | 73.8 | 68.6 | 68.8 | 63.4 |
| DER Network † | 71.3 | 72.9 | – | – | – | – | – | – |
| Stanford AR (relabeling) † | 73.8 | 73.6 | 77.6 | 76.6 | – | – | – | – |
| Iterative Attentive Reader † | 72.6 | 73.3 | – | – | 75.2 | 68.6 | 72.1 | 69.2 |
| EpiReader † | 73.4 | 74.0 | – | – | 75.3 | 69.7 | 71.5 | 67.4 |
| AoA Reader † | 73.1 | 74.4 | – | – | 77.8 | 72.0 | 72.2 | 69.4 |
| ReasoNet † | 72.9 | 74.7 | 77.6 | 76.6 | – | – | – | – |
| NSE † | – | – | – | – | 78.2 | 73.2 | 74.3 | **71.9** |
| BiDAF † | 76.3 | 76.9 | 80.3 | 79.6 | – | – | – | – |
| MemNets (ensemble) † | 66.2 | 69.4 | – | – | – | – | – | – |
| AS Reader (ensemble) † | 73.9 | 75.4 | 78.7 | 77.7 | 76.2 | 71.0 | 71.1 | 68.9 |
| Stanford AR (relabeling,ensemble) † | 77.2 | 77.6 | 80.2 | 79.2 | – | – | – | – |
| Iterative Attentive Reader (ensemble) † | 75.2 | 76.1 | – | – | 76.9 | 72.0 | 74.1 | 71.0 |
| EpiReader (ensemble) † | – | – | – | – | 76.6 | 71.8 | 73.6 | 70.6 |
| AS Reader (+BookTest) † ‡ | – | – | – | – | 80.5 | 76.2 | 83.2 | 80.8 |
| AS Reader (+BookTest,ensemble) † ‡ | – | – | – | – | *82.3* | *78.4* | *85.7* | *83.7* |
| GA-- | 73.0 | 73.8 | 76.7 | 75.7 | 74.9 | 69.0 | 69.0 | 63.9 |
| GA (update $L(w)$) | **77.9** | **77.9** | **81.5** | **80.9** | 76.7 | 70.1 | 69.8 | 67.3 |
| GA (fix $L(w)$) | 77.9 | 77.8 | 80.4 | 79.6 | 77.2 | 71.4 | 71.6 | 68.0 |
| GA (+feature, update $L(w)$) | 77.3 | 76.9 | 80.7 | 80.0 | 77.2 | 73.3 | 73.0 | 69.8 |
| GA (+feature, fix $L(w)$) | 76.7 | 77.4 | 80.0 | 79.3 | **78.5** | **74.9** | **74.4** | 70.7 |

that. This is a common trend in machine learning as model complexity is increased, however we note that a multi-hop architecture is important to achieve a high performance for this task, and provide further evidence for this in the next section.

Lastly, we perform an ablation study for the three components of the GA Reader which were absent in the preprint version (GA Reader--). See Appendix B for more details.

### 4.4 Attention Visualization

To gain an insight into the reading process employed by the model we analyzed the attention distributions at intermediate layers of the reader. Figure 3 shows an example from the validation set of WDW dataset (several more are in the Appendix). In each figure, the left and middle plots visualize attention over the query (equation 5) for candidates in the document after layers 1 & 2 respectively. The right plot shows attention over candidates in the document of cloze placeholder (XXX) in the query at the final layer. The full document, query and correct answer are shown at the bottom.

A generic pattern observed in these examples is that in intermediate layers, candidates in the document (shown along rows) tend to pick out salient tokens in the query which provide clues about the cloze, and in the final layer the candidate with the highest match with these tokens is

Figure 2: Performance in accuracy with and without the Gated-Attention module over different training sizes. $p$-values for an exact one-sided Mcnemar's test are given inside the parentheses for each setting.

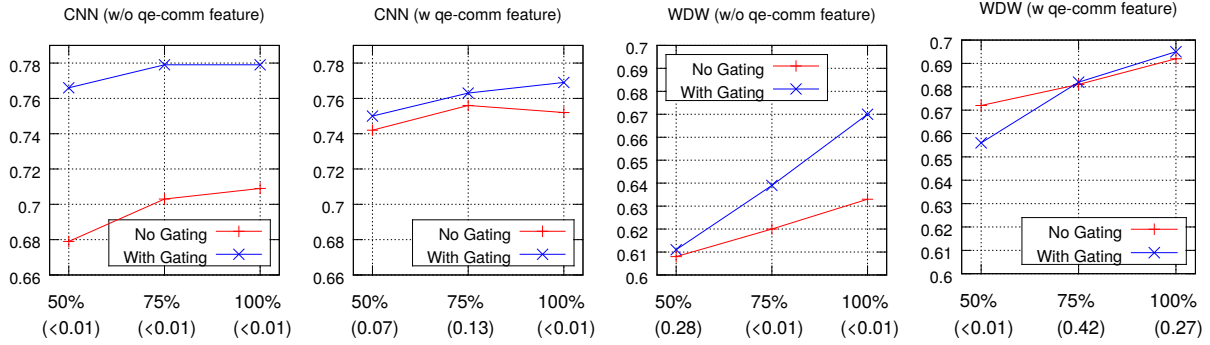

Figure 3: Layer-wise attention visualization of GA Reader trained on WDW-Strict. See text for details.

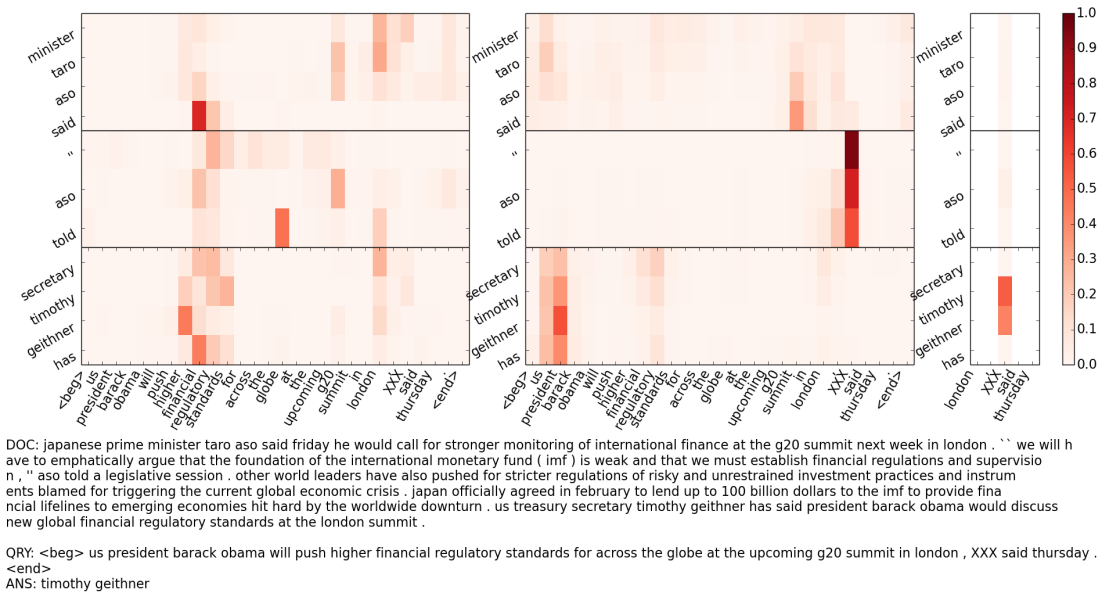

DOC: japanese prime minister taro aso said friday he would call for stronger monitoring of international finance at the g20 summit next week in london . `` we will have to emphatically argue that the foundation of the international monetary fund ( imf ) is weak and that we must establish financial regulations and supervision , '' aso told a legislative session . other world leaders have also pushed for stricter regulations of risky and unrestrained investment practices and instruments blamed for triggering the current global economic crisis . japan officially agreed in february to lend up to 100 billion dollars to the imf to provide financial lifelines to emerging economies hit hard by the worldwide downturn . us treasury secretary timothy geithner has said president barack obama would discuss new global financial regulatory standards at the london summit .

QRY: <beg> us president barack obama will push higher financial regulatory standards for across the globe at the upcoming g20 summit in london , XXX said thursday . <end>
ANS: timothy geithner

selected as the answer. In Figure 3 there is a high attention of the correct answer on `financial regulatory standards` in the first layer, and on `us president` in the second layer. The incorrect answer, in contrast, only attends to one of these aspects, and hence receives a lower score in the final layer despite the n-gram overlap it has with the cloze token in the query. Importantly, different layers tend to focus on different tokens in the query, supporting the hypothesis that the multi-hop architecture of GA Reader is able to combine distinct pieces of information to answer the query.

## 5 Conclusion

We presented the Gated-Attention reader for answering cloze-style questions over documents. The GA reader features a novel multiplicative gating mechanism, combined with a multi-hop architecture. Our model achieves the state-of-the-art performance on several large-scale benchmark datasets with more than 4% improvements over competitive baselines. Our model design is backed up by an ablation study showing statistically significant improvements of using Gated Attention as information filters. We also showed empirically that multiplicative gating is superior to addition and concatenation operations for implementing gated-attentions, though a theoretical justification remains part of future research goals. Analysis of document and query attentions in intermediate layers of the reader further reveals that the model iteratively attends to different aspects of the query to arrive at the final answer. In this paper we have focused on text comprehension, but we believe that the Gated-Attention mechanism may benefit other tasks as well where multiple sources of information interact.

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

## A  Implementation Details

Our model was implemented using the Theano (Theano Development Team, 2016) and Lasagne[4] Python libraries. We used stochastic gradient descent with ADAM updates for optimization, which combines classical momentum and adaptive gradients (Kingma & Ba, 2014). The batch size was 32 and the initial learning rate was $5 \times 10^{-4}$ which was halved every epoch after the second epoch. The same setting is applied to all models and datasets. We also used gradient clipping with a threshold of 10 to stabilize GRU training (Pascanu et al., 2012). We set the number of layers $K$ to be 3 for all experiments. The number of hidden units for the character GRU was set to 50. The remaining two hyperparameters—size of document and query GRUs, and dropout rate—were tuned on the validation set, and their optimal values are shown in Table 5. In general, the optimal GRU size increases and the dropout rate decreases as the corpus size increases.

The word lookup table was initialized with $100d$ GloVe vectors[5] (Pennington et al., 2014) and OOV tokens at test time were assigned unique random vectors. We empirically observed that initializing with pre-trained embeddings gives higher performance compared to random initialization for all

---

[4] https://lasagne.readthedocs.io/en/latest/
[5] http://nlp.stanford.edu/projects/glove/

datasets. Furthermore, for smaller datasets (WDW and CBT) we found that fixing these embeddings to their pretrained values led to higher test performance, possibly since it avoids overfitting. We do not use the character composition model for CNN and Daily Mail, since their entities (and hence candidate answers) are anonymized to generic tokens. For other datasets the character lookup table was randomly initialized with $25d$ vectors. All other parameters were initialized to their default values as specified in the Lasagne library.

## B  Ablation Study for Model Components

Table 6 shows accuracy on WDW by removing one component at a time.

Table 6: Ablation study on WDW dataset, without using the qe-comm feature and with fixed $L(w)$. Results marked with † are cf (Onishi et al., 2016).

| Model | Accuracy | |
|---|---|---|
| | Val | Test |
| GA | **68.3** | **68.0** |
| −char | 66.9 | 66.9 |
| −token-attentions (eq. 5) | 65.7 | 65.0 |
| −glove, +corpus | 64.0 | 62.5 |
| GA--† | – | 57 |

The steepest reduction is observed when we replace pretrained GloVe vectors with those pretrained on the corpus itself. GloVe vectors were trained on a large corpus of about 6 billion tokens (Pennington et al., 2014), and provide an important source of prior knowledge for the model. Note that the strongest baseline on WDW, NSE (Munkhdalai & Yu, 2016b), also uses pretrained GloVe vectors, hence the comparison is fair in that respect. Next, we observe a substantial drop when removing token-specific attentions over the query in the GA module, which allow gating individual tokens in the document only by parts of the query relevant to that token rather than the overall query representation. Finally, removing the character embeddings, which were only used for WDW and CBT, leads to a reduction of about 1% in the performance.

## C  Attention Plots

Table 4: Dataset statistics.

|  | CNN | Daily Mail | CBT-NE | CBT-CN | WDW-Strict | WDW-Relaxed |
|---|---|---|---|---|---|---|
| # train | 380,298 | 879,450 | 108,719 | 120,769 | 127,786 | 185,978 |
| # validation | 3,924 | 64,835 | 2,000 | 2,000 | 10,000 | 10,000 |
| # test | 3,198 | 53,182 | 2,500 | 2,500 | 10,000 | 10,000 |
| # vocab | 118,497 | 208,045 | 53,063 | 53,185 | 347,406 | 308,602 |
| max doc length | 2,000 | 2,000 | 1,338 | 1,338 | 3,085 | 3,085 |

Table 5: Hyperparameter settings for each dataset. dim() indicates hidden state size of GRU.

| Hyperparameter | CNN | Daily Mail | CBT-NE | CBT-CN | WDW-Strict | WDW-Relaxed |
|---|---|---|---|---|---|---|
| Dropout | 0.2 | 0.1 | 0.4 | 0.4 | 0.3 | 0.3 |
| $\dim(\overleftrightarrow{\text{GRU}}_*)$ | 256 | 256 | 128 | 128 | 128 | 128 |

Figure 4: Layer-wise attention visualization of GA Reader trained on WDW-Strict. See text for details.

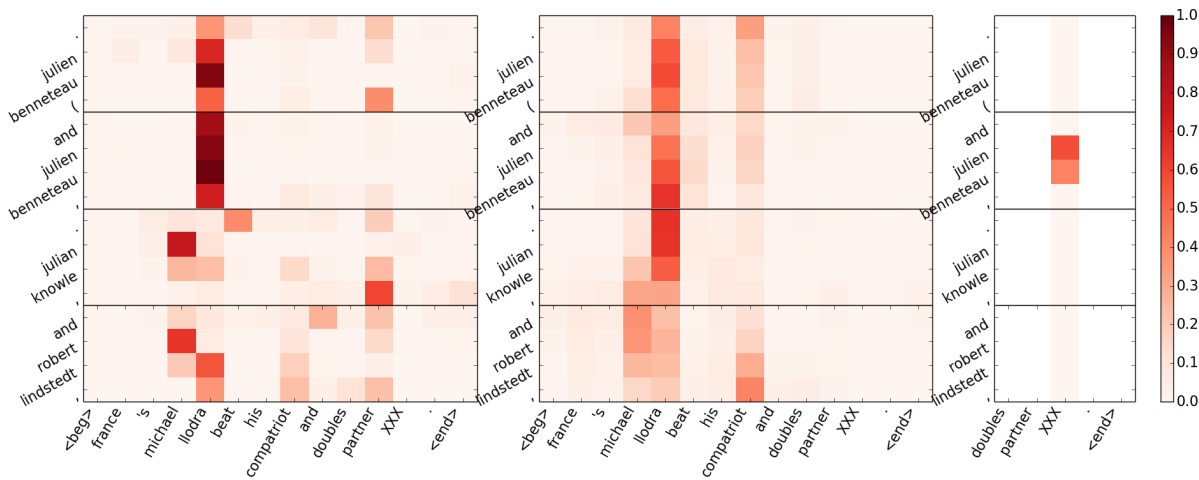

DOC: result sunday from the open 13 , a ( euro ) 512,750 ( $ 697,400 ) atp world tour indoor hardcourt event at palais des sports ( seedings in parentheses ) : singles final michael llodra , france , def . julien benneteau ( 8 ) , france , 6-3 , 6-4. doubles final michael llodra and julien benneteau , france ( 2 ) , def . julian knowle , austria and robert lindstedt , sweden ( 1 ) , 6-4 , 6-3 .

QRY: <beg> france 's michael llodra beat his compatriot and doubles partner XXX . <end>
ANS: julien benneteau

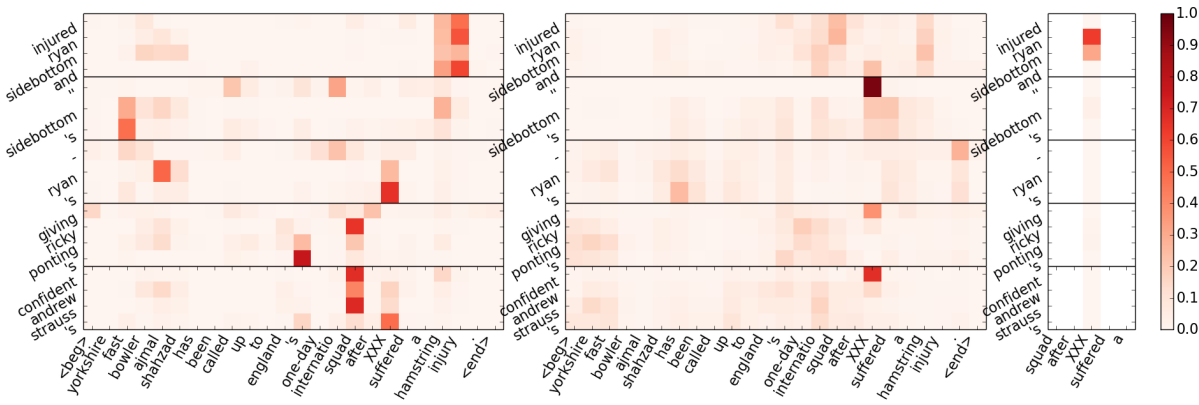

DOC: england pace bowler ajmal shahzad has warned australia that his `` fearless '' team- mates are ready to deal their rivals a psychological blow in the forthcoming one-day series . shahzad has been called into england 's squad for five matches against australia as cover for the injured ryan sidebottom and he senses an extremely positive mood in the dressing room . the 24-year-old admits england are desperate to warm up for the ashes in australia later this year by giving ricky ponting 's men another beating after defeating them in the recent icc world twenty20 final in the caribbean . and shahzad , the first british-born asian to play for yorkshire , is confident andrew strauss 's side wo n't back down against the ultra-aggressive australians . `` it will be a step up for the lads . but everybody is focused and ready for it , '' shahzad said . `` there 's no fear , no nerves - and it 's nice to be part of that kind of dressing room . '' sidebottom 's hamstring problem has given shahzad his opportunity in the one-day squad and he is determined to seize the opportunity to stake his claim for a permanent place by impressing against the australians . `` it 's just nice to be called up , and i hope if my chance comes i can grasp it with both hands , '' he said . `` a lot 's happened to me in the last six months . if the chance comes - ryan 's got a niggle , but i do n't think it 's too bad - i 've just got to put in a decent performance and have my name in the hat for the games to come . `` but it 's just nice to be involved , and know you 're there or thereabouts . i just hope i can get the nod . ''

QRY: <beg> yorkshire fast bowler ajmal shahzad has been called up to england 's one-day international squad after XXX suffered a hamstring injury . <end>
ANS: ryan sidebottom

Figure 5: Layer-wise attention visualization of GA Reader trained on WDW-Strict. See text for details.

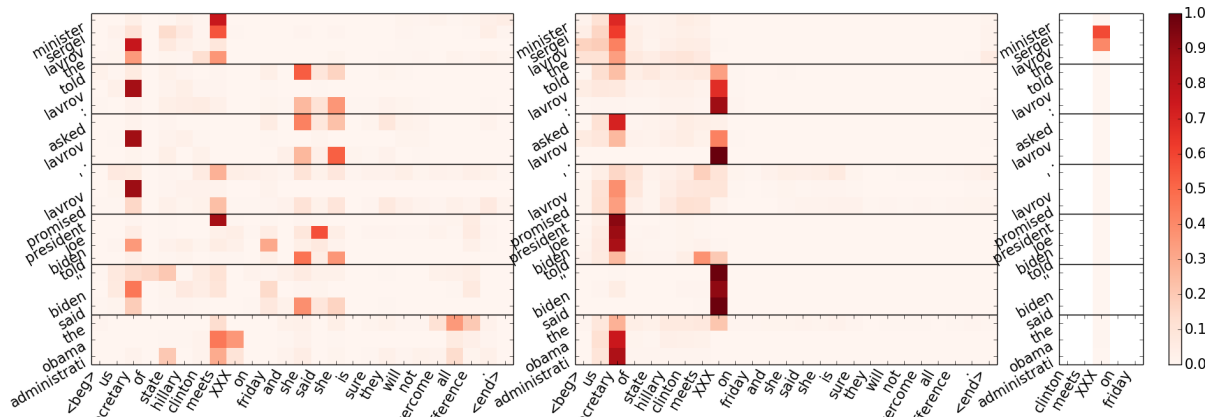

DOC: us secretary of state hillary clinton sought to break the ice with her russian counterpart on friday by handing him a fake `` reset '' button -- or at least that it was what it was supposed to be . clinton handed russian foreign minister sergei lavrov the button wrapped in a ribbon as they began their first meeting in a luxury hotel in geneva . earlier this year , us vice president joe biden told foreign leaders at an international security conference in munich , southern germany , that the obama administration wanted to improve ties with moscow . `` it is time to press the reset button and to revisit the many areas where we can and should work together , '' biden said . as she proffered the red plastic button , clinton told lavrov : `` we want to reset our relationship . and so we will do it together , '' she said , laughing . but the button also bore a russian word that was meant to translate as `` reset '' . `` we worked hard to get the right russian word . do you think we got it ? '' clinton asked lavrov . `` you got it wrong , '' he responded as they both laughed . `` it should be 'perezagrouzka ' ( the russian word for 'reset ' ) , '' the russian foreign minister pointed out . `` this says , peregruzka , which means overcharged . '' `` we wo n't let you do that to us , '' clinton replied . russian speakers indicated that the mistaken word was better translated as 'overload . ' lavrov promised to keep it on his desk .

QRY: <beg> us secretary of state hillary clinton meets XXX on friday and she said she is sure they will not overcome all differences . <end>
ANS: sergei lavrov

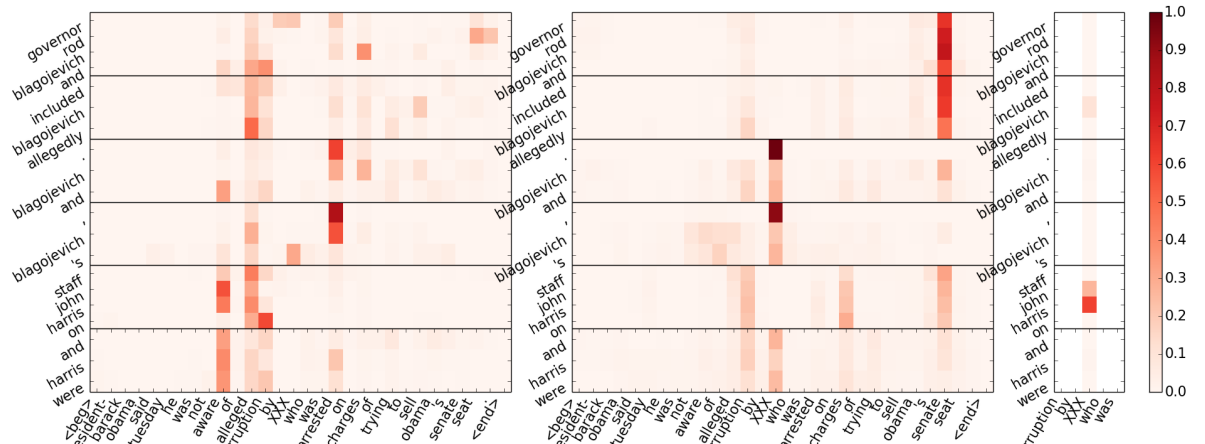

DOC: illinois governor arrested on corruption charges chicago , dec. 9 ( xinhua ) -- u.s. federal prosecutors on tuesday arrested illinois governor rod blagojevich and his chief of staff john harris on corruption charges . the two were accused of a wide-ranging criminal conspiracy that included blagojevich allegedly conspiring to sell or trade the senate seat left vacant by president-elect barack obama in exchange for financial benefits for his wife and himself . the governor was also charged with obtaining campaign contributions in exchange for other official actions . blagojevich and harris were arrested simultaneously at their homes at about 6:15 a.m. , according to the fbi . they were transported to fbi headquarters in chicago , where they remained at 9 a.m. however , blagojevich 's spokesman said he did not know the development .

QRY: <beg> president-elect barack obama said tuesday he was not aware of alleged corruption by XXX who was arrested on charges of trying to sell obama 's senate seat . <end>
ANS: rod blagojevich

Figure 6: Layer-wise attention visualization of GA Reader trained on WDW-Strict. See text for details.

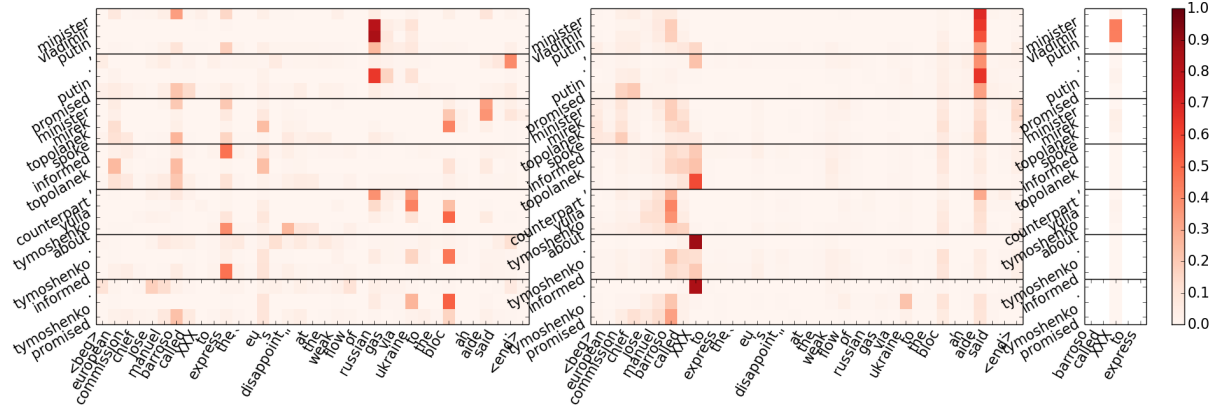

DOC: european union officials complained on tuesday about the lack of gas flow from russia through ukraine to europe after russia resumed early gas supplies under a three- way deal signed on the previous day . european commission president jose manuel barroso spoke by phone to russian prime minister vladimir putin , express ing disappointment over the lack of natural gas flowing to europe . eu monitors on the ground reported that only very little gas is flowing through the pipeline s . barroso voiced his `` disappointment with both the level of gas flowing to europe '' and the lack of access `` of our monitors to dispatch centers , '' acco rding to his aide . putin promised him to take a look into what he complained . on the same day , czech prime minister mirek topolanek spoke on the phone to his ukrainian counterpart yulia tymoshenko about the matter , said a press release from the czech eu presidency . tymoshenko informed topolanek , who asked about t he causes and circumstances of the delay in supplies , of some technical difficulties , saying that more specifically the pressure of gas arriving from the russ ia is too low . the czech prime minister recommended her to immediately contact the eurogaz experts who are ready to assist ukraine with technical problems . ty moshenko promised to act on this offer . russia reopened taps tuesday morning to let gas flow to europe via ukraine after cutting off gas supplies to europe on wednesday amid a pricing dispute with ukraine . the cutoff left a number of european countries in lack of heating gas amid freezing weather .

QRY: <beg> european commission chief jose manuel barroso called XXX to express the `` eu 's disappointment '' at the weak flow of russian gas via ukraine to the bloc , an aide said . <end>
ANS: vladimir putin

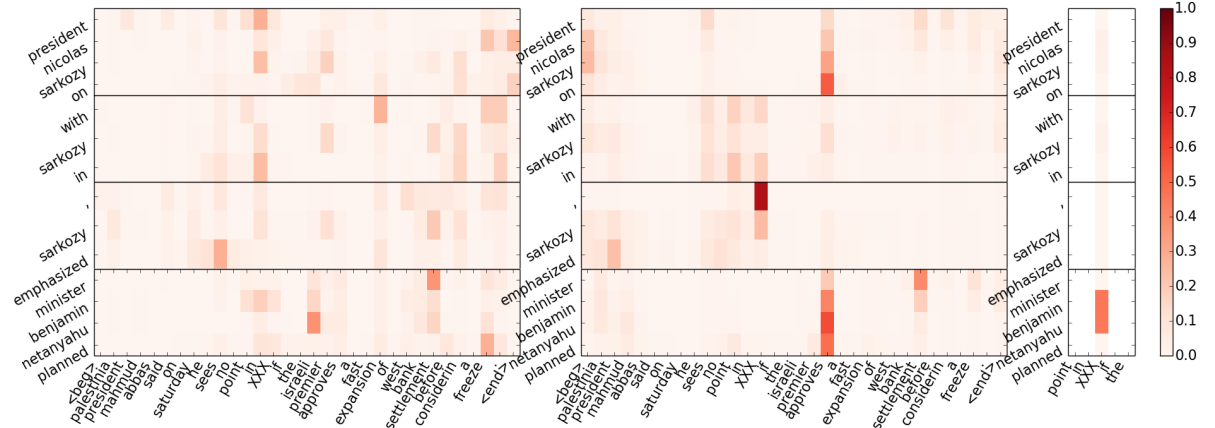

DOC: president of the palestinian authority mahmud abbas held talks with french president nicolas sarkozy on friday , and slammed the israeli plan of constructing mo re settlement buildings on the west bank as `` unacceptable . '' `` that is not acceptable , '' said abbas after meeting with sarkozy in the elysee palace . it was reported that israeli prime minister benjamin netanyahu planned to approval the construction of new home buildings on the west bank before considering a fre eze on settlement activities . `` we want a freeze on settlement and the launch of negotiations on the final phase of it , '' abbas said . `` this was the main subject of our talks . '' according to a statement from the elysee palace , talks between the two leaders were aimed at starting again the peace process within the palestinian territories , as well as discussing regional issues . during the meeting , sarkozy emphasized the urgency of the resumption of a negotiation pro cess between israel and palestine . this has been abbas ' third official visit to france since 2007 .

QRY: <beg> palestinian president mahmud abbas said on saturday he sees no point in XXX if the israeli premier approves a fast expansion of west bank settlements befo re considering a freeze . <end>
ANS: benjamin netanyahu

Figure 7: Layer-wise attention visualization of GA Reader trained on WDW-Strict. See text for details.

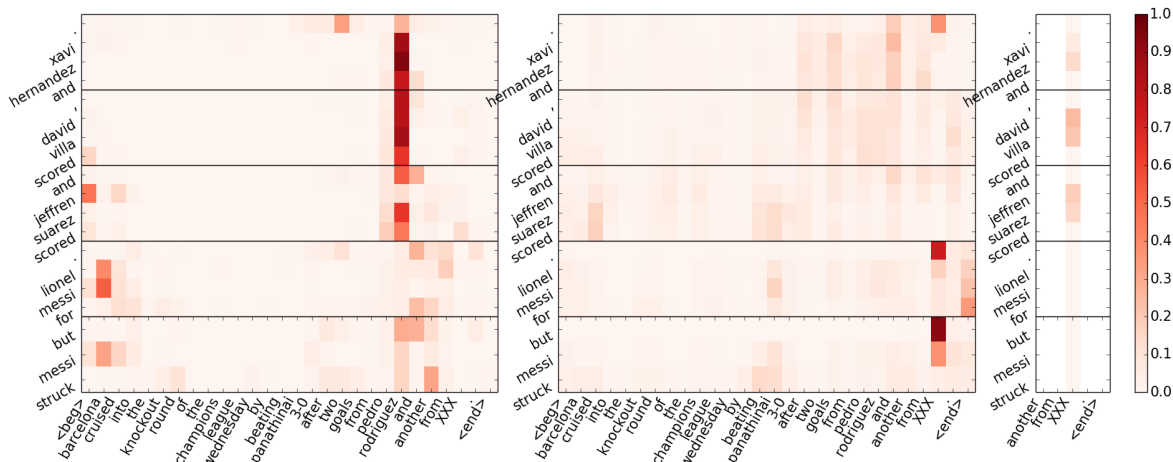

DOC: if there was a shred of doubt where the world cup was built and won for spain this year , it was removed monday night when barcelona destroyed real madrid , 5-0 . in teeming rain , host barcelona simply outplayed real , which led the spanish league until monday . barcelona 's lineup contained eight home-bred players , seven of them world champions . xavi hernandez and pedro rodriguez scored , david villa scored twice , and jeffren suarez scored the fifth as a substitute . the loss ended madrid 's 26-game unbeaten streak . lionel messi for once did not score . but messi struck a post , and he was involved in three of the goals . indeed , by taking a deeper role and taking considerable brutish tackles , he epitomized barcelona 's collective will to work for one another .

QRY: <beg> barcelona cruised into the knockout round of the champions league wednesday by beating panathinaikos 3-0 after two goals from pedro rodriguez and another from XXX . <end>
ANS: lionel messi

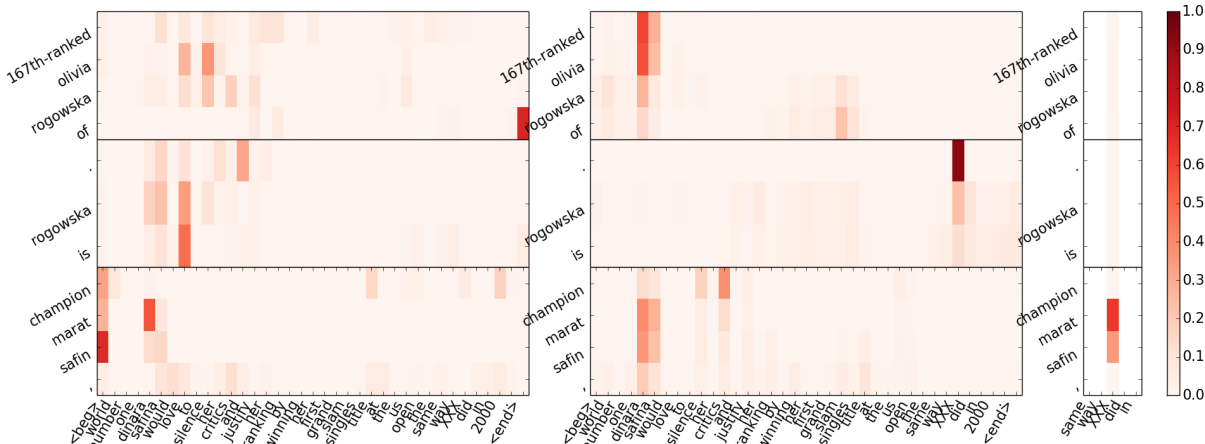

DOC: dinara safina has barely avoided becoming the first no . 1-seeded woman to lose in the first round at the u.s. open . safina overcame 11 double-faults and 48 unforced errors to come back and beat 167th-ranked olivia rogowska of australia 6-7 ( 5 ) , 6-2 , 6-4 in arthur ashe stadium on tuesday . safina , the younger sister of two-time major champion marat safin , moved up to no . 1 in the rankings in april -- and is assured of staying there no matter what happens at flushing meadows . the russian reached the finals at the australian open and french open this year , losing both . rogowska is 18 , received a wild-card invitation into the u.s. open and has won one grand slam match . she never has defeated anyone ranked better than 47th .

QRY: <beg> world number one dinara safina would love to silence her critics and justify her ranking by winning her first grand slam singles title at the us open the same way XXX did in 2000 . <end>
ANS: marat safin

