# Peer review of "Gated-Attention Readers for Text Comprehension"

_ACL 2017 — decision unknown_

[Official Review · Reviewer 1 · rating 4 · confidence 4]
soundness 3 · originality 4 · clarity 3 · impact 4 · substance 4 · appropriateness 5 · meaningful comparison 4 · presentation format Poster

This paper presents a gated attention mechanism for machine reading. 
A key idea is to extend Attention Sum Reader (Kadlec et al. 2016) to multi-hop
reasoning by fine-grained gated filter. 
It's interesting and intuitive for machine reading. 
I like the idea along with significant improvement on benchmark datasets, but
also have major concerns to get it published in ACL.

- The proposed GA mechanism looks promising, but not enough to convince the
importance of this technique over other state-of-the-art systems, because
engineering tricks presented 3.1.4 boost a lot on accuracy and are blended in
the result.

- Incomplete bibliography: Nearly all published work in reference section
refers arxiv preprint version. 
This makes me (and future readers) suspicious if this work thoroughly compares
with prior work. Please make them complete if the published version is
available. 

- Result from unpublished work (GA): GA baseline in table 1 and 3 is mentioned
as previous work that is unpublished preprint. 
I don't think this is necessary at all. Alternately, I would like the author to
replace it with vanilla GA (or variant of the proposed model for baseline). 
It doesn't make sense that result from the preprint which will end up being the
same as this ACL submission is presented in the same manuscript. 
For fair blind-review, I didn't search on arvix archive though.

- Conflict on table 1 and 2: GA-- (table 1) is the same as K=1(AS) in table 2,
and GA (fix L(w)) is for K=3 in table 2. 
Does this mean that GA-- is actually AS Reader? 
It's not clear that GA-- is re-implementation of AS. 
I assumed K=1 (AS) in table 2 uses also GloVe initialization and
token-attention, but it doesn't seem in GA--. 

- I wish the proposed method compared with prior work in related work section
(i.e. what's differ from related work).

- Fig 2 shows benefit of gated attention (which translates multi-hop
architecture), and it's very impressive. It would be great to see any
qualitative example with comparison.

[Official Review · Reviewer 2 · rating 3 · confidence 3]
soundness 3 · originality 4 · clarity 4 · impact 3 · substance 3 · appropriateness 4 · meaningful comparison 3 · presentation format Oral Presentation

This paper presents an interesting model for reading comprehension, by
depicting the multiplicative interactions between the query and local
information around a word in a document, and the authors proposed a new
gated-attention strategy to characterize the relationship. The work is quite
solid, with almost state of art result on the whole four cloze-style datasets
achieved. Some of the further improvement can be helpful for the similar tasks.


Nevertheless, I have some concerns on the following aspect:

1. The authors have referred many papers from arXiv, but I think some really
related works are not included. Such as the works from Caiming Xiong, et al.
https://openreview.net/pdf?id=rJeKjwvclx and the work form Shuohang Wang, et
al. https://openreview.net/pdf?id=B1-q5Pqxl . Both of them concentrated on
enhancing the attention operation to modeling the interaction between documents
and queries. Although these works are not evaluated on the cloze-style corpus
but the SQuAD, an experimental or fundamental comparison may be necessary.

2. There have been some studies that adopts attention mechanism or its variants
specially designed for the Reading Comprehension tasks, and the work actually
share the similar ideas with this paper. My suggestion is to conduct some
comparisons with such work to enhance the experiments of this paper.

[Official Review · Reviewer 3 · rating 4 · confidence 3]
soundness 3 · originality 4 · clarity 5 · impact 5 · substance 4 · appropriateness 5 · meaningful comparison 5 · presentation format Oral Presentation

- Strengths:

* Paper is very well-written and every aspect of the model is well-motivated
and clearly explained.
* The authors have extensively covered the previous work in the area.
* The approach achieves state-of-the-art results across several text
comprehension data sets. In addition, the experimental evaluation is very
thorough.

- Weaknesses:

* Different variants of the model achieve state-of-the-art performance across
various data sets. However, the authors do provide an explanation for this
(i.e. size of data set and text anonymization patterns).

- General Discussion:

The paper describes an approach to text comprehension which uses gated
attention modules to achieve state-of-the-art performance. Compared to previous
attention mechanisms, the gated attention reader uses the query embedding and
makes multiple passes (multi-hop architecture) over the document and applies
multiplicative updates to the document token vectors before finally producing a
classification output regarding the answer. This technique somewhat mirrors how
humans solve text comprehension problems. Results show that the approach
performs well on large data sets such as CNN and Daily Mail. For the CBT data
set, some additional feature engineering is needed to achieve state-of-the-art
performance. 

Overall, the paper is very well-written and model is novel and well-motivated.
Furthermore, the approach achieves state-of-the-art performance on several data
sets. 

I had only minor issues with the evaluation. The experimental results section
does not mention whether the improvements (e.g. in Table 3) are statistically
significant and if so, which test was used and what was the p-value. Also I
couldn't find an explanation for the performance on CBT-CN data set where the
validation performance is superior to NSE but test performance is significantly
worse.